

# Inter-annual variability of mean sea level and its sensitivity to wind climate in an inter-tidal basin

Theo Gerkema[1] and Matias Duran-Matute[2,1]

[1]Department for Estuarine and Delta Systems, NIOZ Netherlands Institute for Sea Research, and Utrecht University, PO Box 140, 4400 AC Yerseke, The Netherlands
[2]Department of Applied Physics, Technical University Eindhoven, PO Box 513, 5600 MB Eindhoven, The Netherlands

*Correspondence to:* Theo Gerkema (gerk@nioz.nl)

**Abstract.** The relation between the annual wind records from a weather station and annual mean sea level in an inter-tidal basin, the Dutch Wadden Sea, is examined. Only more recent, homogeneous wind records are used, covering the past two decades. It is demonstrated that even such a relatively short record is sufficient to find a convincing relation. The inter-annual variability of mean sea level can already be largely explained by the west-east component of the net wind energy vector, with some

further improvement if one also includes the south-north component and theannual mean atmospheric pressure. For different tide-gauge stations in the Dutch Wadden Sea and along the coast, we find the same qualitative characteristics, although the precise values of the correlations vary. Correcting observed values of annual mean level for meteorological factors reduces the margin of error (expressed as 95%-confidence interval) by about a factor of three in the trends of the 20-year sea level record. Model results illustrate the regional variability in annual mean sea level and its inter-annual variability. The sensitivity on wind

direction varies spatially even on a small scale like the Dutch Wadden Sea. This study also implies that climatic changes in the strength of winds from a specific direction may affect local annual mean sea level quite significantly.

## 1   Introduction

Changes in relative mean sea level affect coastal areas in various ways, such as altering the risk of flooding, the evolution of barrier island systems, or the development of salt marshes, as reviewed by FitzGerald et al. (2008). Trends in these changes

are partly masked by variability on time scales from days to decades. Some of this variability, for instance due to wind-waves and tides (with the exception of long-period tides), is easily averaged out. In contrast, inter-annual variability is found to be irregular and large, of the order of a few decimeters, as is evident from tide gauge records around the world (see, for instance, Zervas (2009) for examples at the US coast or Figure 2). This is why the climatic trend, typically of a few millimeters per year, can only be reliably identified by examining a record that is long enough to outweigh the inter-annual and decadal variabilities.

An estimate of the required length was derived by Zervas (2009): the 95%-confidence interval is nearly $\pm 3$ mm/yr for a 20-year record, but drops quickly for longer ones; for a 60-year record, the confidence interval is already below $\pm 0.5$ mm/yr.

The meteorological factors of wind and atmospheric pressure have been identified as possible causes of inter-annual and decadal variability of mean sea level (Stammer et al., 2013). For example, on a decadal time scale, the steep rise of annual mean sea level in the western Pacific Ocean was traced back to a strengthening of Trade Winds in a modelling study by Merrifield



and Maltrud (2011). Sea level variations in the Indo-Pacific region were shown to be connected with decadal variability in wind stress (Lee and McPhaden, 2008). Correlations have been reported between annual mean atmospheric pressure and annual mean sea level, such as for the southwestern British coast (Pugh, 2004) and the Norwegian coast (Richter et al., 2012). Dangendorf et al. (2013) examined trends in annual mean sea level as well as trends for individual seasons (i.e., quarterly mean values),

which turned out to differ significantly due to wind climate (we come back to this phenomenon in Section 2.1). With a view to long-term coastal protection, de Ronde et al. (2014) reported on trends in sea level rise along the Dutch coast, with and without correcting for meteorological factors and other effects (such as the 18.6 year cycle). They analyzed the period 1970-2012 and found a strong effect of wind on annual mean sea level.

In this study, we elaborate on these results and examine the regional variability of the sensitivity to wind climate. We focus

on the inter-tidal area of the Dutch Wadden Sea, which offers an interesting case because of the complex morphology and shallowness, and hence, a complex response to wind in the circulation and sea level (Duran-Matute et al., 2014, 2016). We include annual mean atmospheric pressure along with the annual characteristics of the wind climate. The aim is to relate inter-annual variability of mean sea level with data from meteorological records and determine the regional variations in this relation. Using meteorological records involves the challenge of finding reliable and consistent long-term time series of wind speed and

direction. For example, some wind records from weather stations in the Netherlands (the case studied in this paper) date back to the early 20th century, but they are unsuitable for trend analysis because of inhomogeneities in the record (see Section 2.2). For this reason, we will use only more recent wind records, from the past two decades. The question then is whether such a relatively short record is sufficient to find a convincing relation with annual mean sea level. It is the purpose of this paper to demonstrate that the answer is positive and to suggest a method to find and exploit such a relation. An alternative would be to

use atmospheric reanalysis model data; examples are the studies by Dangendorf et al. (2013) and Baart et al. (2014).

Using annual mean wind energy (split into sectorial or vectorial directions) and annual mean atmospheric pressure to examine their relation to annual mean sea level, assumes an underlying linearity in the cause-effect relation. While at shorter time scales (hours, days) there is a direct mechanistic relation between wind forcing or atmospheric pressure and sea level (as demonstrated by the accuracy of hydrodynamical models, e.g., Zijl et al. (2013)), it is not a priori clear that this relation is

carried over to their annual mean values. The validity of this approach will become evident from the results.

In Section 2, we present the data records and methods used in this paper, both for annual mean sea level and the wind climate. In Section 3, we examine the correlations between the two and present annual mean sea levels that are corrected for meteorological effects. In Section 4, the regional patterns characterizing inter-annual variability of annual mean sea level are illustrated by means of numerical model results for the Dutch Wadden Sea. Finally, we discuss unresolved issues and

summarize our findings in Sections 5 and 6.



## 2 Data records and methodology

We examine relative sea level changes at the Dutch coast, with a focus on three tide gauges in the Dutch Wadden Sea (from northeast westward: Delfzijl, Harlingen and Den Helder), in relation to the wind climate from the record at weather station Vlieland. All relevant locations are indicated on the map in Figure 1.

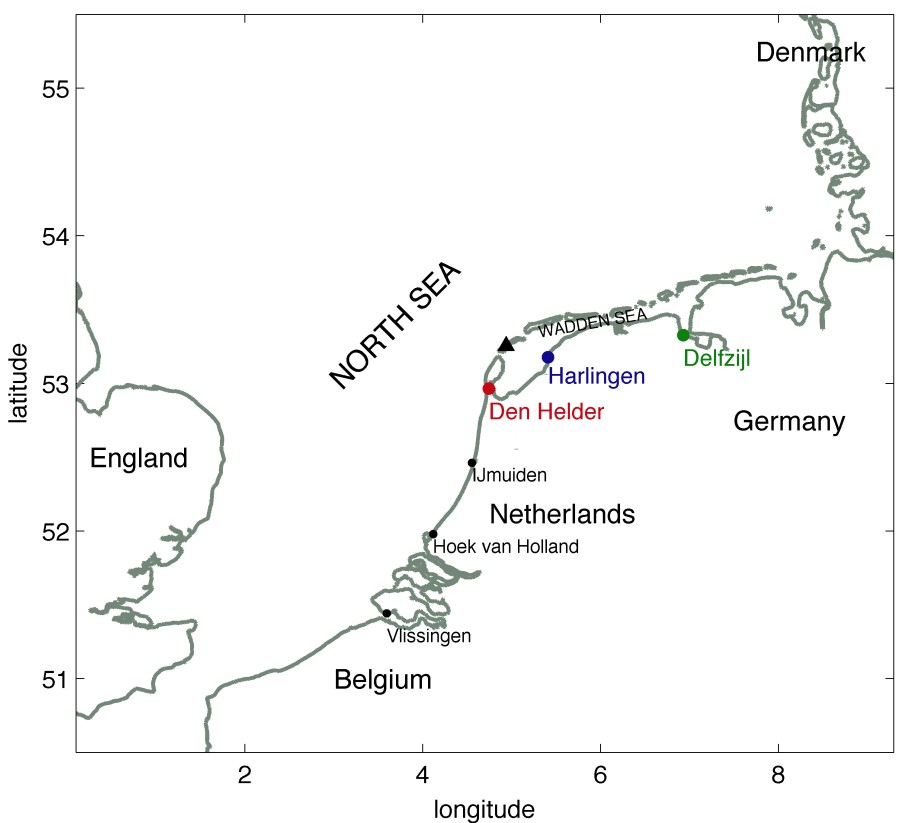

**Figure 1.** A map depicting the tide gauge stations analyzed in this paper: Delfzijl (green), Harlingen (blue) and Den Helder (red). Other stations (IJmuiden, Hoek van Holland and Vlissingen) are only briefly discussed. The location of weather station Vlieland is indicated by a black triangle.

### 2.1 Sea level variations

At the Permanent Service for Mean Sea Level (PSMSL), data from tide gauges along the Dutch coast (supplied by Rijkswaterstaat) is corrected to the Revised Local Reference (Woodworth and Player, 2003). In Figure 2, we show the annual mean values of sea level for the period 1966-2015, for three tide gauges in the Dutch Wadden Sea. Trends, plotted in solid lines, are based on this 50-year period.





We obtained the trends from a least squares fit. The 95%-confidence interval on the slope is indicated in the legend (see Montgomery and Runger (2003) for a description of the method of calculation.) To test the robustness of these results, we alternatively calculated the trends by using the Theil-Sen method (e.g. Sprent, 1993), which has the advantage of being less sensitive to outliers than linear regression. The difference between the methods was found to be slight: 0.1 mm/yr at most.

Hereafter, we opted for linear regression because the associated method of obtaining confidence intervals is more firmly established.

The trends differ between the stations but not beyond the margin of uncertainty, which is indicated in the legend of Figure 2. A longer record is needed to sufficiently reduce the confidence intervals and ascertain a difference in trends, as demonstrated by de Ronde et al. (2014) and seen in the results from a 100-yrs record shown in Table 1.

Our focus in this paper is on the last twenty years, i.e. the period 1996-2015 (indicated in solid lines in Figure 2). It is of great interest to know whether the trend from the past half century has continued during the last two decades, or whether there is an acceleration or deceleration in the rise. However, the large inter-annual variability (see especially the extremely low levels in 1996) prevents us from drawing any conclusion on the most recent trend. For the period 1996-2015, taken in isolation, the 95%-confidence intervals would be as large as ±3.2 mm/yr (the average for the three stations marked in color in Figure

1), which is in close agreement with the estimate by Zervas (2009) for a 20-year period. This very wide confidence interval precludes a meaningful estimate of a trend.

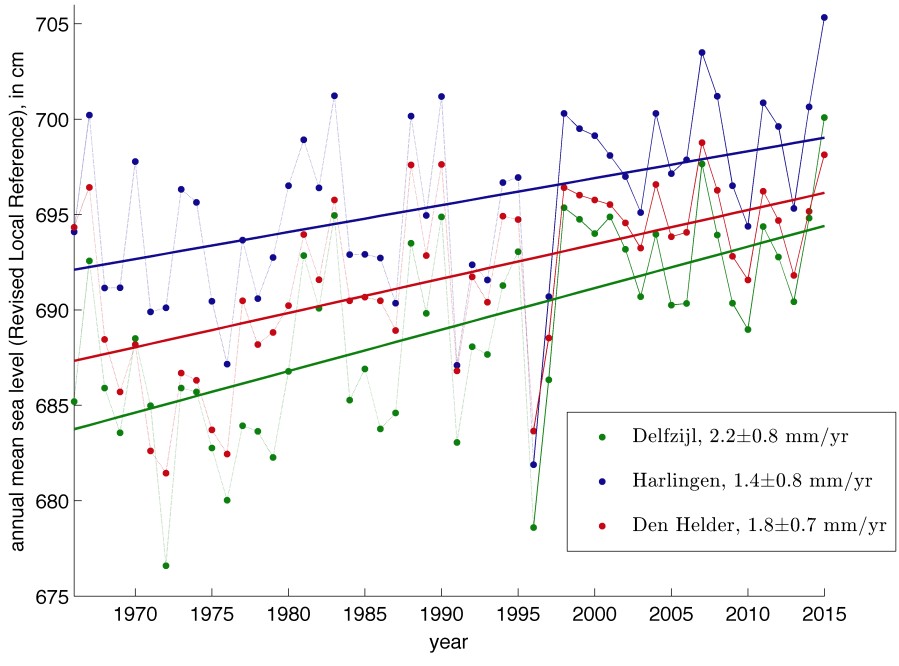

**Figure 2.** Annual mean sea level at three tide gauges in the Dutch Wadden Sea, during the past half century. The record of the last 20 years is indicated in thin solid lines. The 50-year trend (least squares fit) is plotted in thick solid lines and listed in the legend, together with the 95%-confidence interval.



Instead of looking at annual mean values of sea level, we may select a sub-interval, for example a particular month or quarter. For Cuxhaven in the German Bight, Dangendorf et al. (2013) demonstrated that the inter-annual variability in mean values for the second and third quarters is markedly lower than those for the first and fourth quarters – a reflection of the difference in wind intensities between the summer and winter half-years (cf. Figure 4a). Hence the summer half-year offers a more suitable

starting point for deducing long-term trends than the full year, since the noise of inter-annual variability is weaker. However, Dangendorf et al. (2013) also demonstrated that the trends are not the same for different quarters. As a consequence, selecting the summer half-year does not necessarily reveal the trend of annual mean sea level (listed in the first column of Table 1; these values agree with previously calculated trends by Wahl et al. (2013)). This is not only true for the location examined by Dangendorf et al. (2013), but it also holds for the stations along the Dutch coast. We show an example in Figure 3 and list

values for other stations in Table 1. During the past hundred years, mean winter half-year levels (October-March, combining the fourth and first quarters from successive years) have risen more strongly than mean summer half-year levels (April-September). This holds for all stations; the most conspicuous example is Harlingen, where mean winter levels rose nearly twice as fast as mean summer levels. This phenomenon seems to have been overlooked in projections of risks of flooding, where trends of annual-mean values are the default reference. The distinction is relevant because severe storms (in Northwest Europe) mostly

occur in the winter half-year, when the background mean sea level is already higher than the annual mean and moreover has been rising more rapidly, as Table 1 shows.

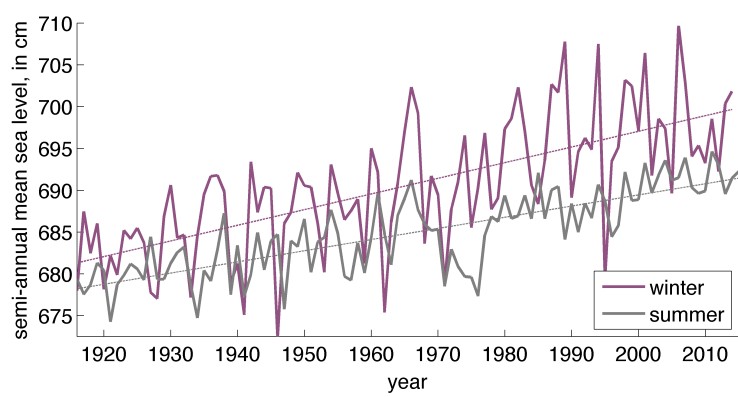

**Figure 3.** Evolution of winter and summer half-year mean sea levels at Den Helder and corresponding trends from linear regression. See also Table 1 for the slope and confidence interval of the trends.

Besides the problem of a difference in trend between summer half-year and full-year, the reduction in confidence intervals gained by selecting the summer half-year is actually modest (see Table 1). Hereafter, we therefore focus on annual mean values, i.e. derived from full-year data.

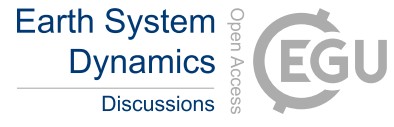

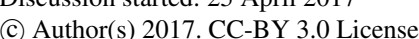


| station | annual-mean | summer-mean | winter-mean |
|---|---|---|---|
| Delfzijl | $1.95 \pm 0.26$ | $1.65 \pm 0.21$ | $2.28 \pm 0.51$ |
| Harlingen | $1.27 \pm 0.27$ | $0.88 \pm 0.24$ | $1.69 \pm 0.50$ |
| Den Helder | $1.59 \pm 0.23$ | $1.34 \pm 0.21$ | $1.87 \pm 0.41$ |
| IJmuiden | $2.15 \pm 0.24$ | $1.87 \pm 0.23$ | $2.48 \pm 0.40$ |
| Hoek van Holland | $2.36 \pm 0.23$ | $2.13 \pm 0.21$ | $2.64 \pm 0.37$ |
| Vlissingen | $2.07 \pm 0.20$ | $1.90 \pm 0.19$ | $2.25 \pm 0.32$ |

**Table 1.** Trends in sea level rise and 95%-confidence intervals (all in mm/yr) for annual and seasonal half-year means. [Derived from 100-yr records, 1916-2015, supplied by Rijkswaterstaat and PSMSL.]

## 2.2 Wind record

We analyze records of wind data from weather station Vlieland (KNMI station 242), which is located on a large sand flat. There are no obstacles in its immediate vicinity, and it lies well exposed to winds from all angles. The data is publicly available from a portal of the KNMI (Royal Netherlands Meteorological Institute). Records of wind speed and direction from KNMI weather

stations span in some cases more than a century. However, from time to time, changes have occurred in the measurement techniques, which may have produced inhomogeneities in the time series. This involves, for instance, changes in the height or location of the instruments, a replacement of instruments, changes in surrounding vegetation or buildings, or changes in protocol, as documented by Verkaik (2001). Thus, the KNMI data comes with the caveat that the "series are not suitable for trend analysis". For example, the weather station Kooy/Den Helder (used by de Ronde et al. (2014)) has been subject to some

relocations, which has impaired the homogeneity of the record. To avoid these problems, we restrict ourselves to data from the 20-year period 1996-2015 recorded by the automatic weather station on Vlieland, without apparent inhomogeneities (apart from a few gaps, discussed below).

Hourly values of wind speed $W$ and direction $D$ are used. They are defined as the mean speed and direction during the last 10-minute interval of the preceding hour and are labeled with hourly interval index $i$.

We divide the wind direction into eight sectors, labeled with $n = 1, \ldots, 8$, respectively: northerly (N), northeasterly (NE), easterly (E), southeasterly (SE), southerly (S), southwesterly (SW), westerly (W), northwesterly (NW). This is the direction *from* which the wind blows.

To characterize the wind climate, we will use wind energy, but other quantities could be used as well. Another natural choice would be the wind stress. However, one then has to adopt a formulation for the drag coefficient in terms of wind strength,

which should be valid for the entire range from breezes to hurricanes. A simple form of the drag coefficient (e.g. Guan and Xie, 2004) would involve a linear dependence on wind speed $W$, implying a cubic power in the stress. By using energy, we get that power straight away without having to enter the uncertain territory of how to define the drag coefficient. We carried out tests and found that the results are actually not very sensitive to the choice of the power. This agrees with the finding by Richter et al. (2012) that it is immaterial whether one uses wind speed or stress.





Hourly wind speed (as defined above) has a certain magnitude $W_{n,i}$, with sectorial direction $n$. The kinetic energy $E_n$ of wind crossing a vertical plane area $A$ can be written:

$$E_{n,i} = \frac{1}{2} m_{n,i} W_{n,i}^2 = \frac{1}{2} \rho V_{n,i} W_{n,i}^2 = \frac{1}{2} \rho A \Delta t \, W_{n,i}^3,$$

with mass $m$ and volume $V$, which equals the area $A$ times the length $W \Delta t$ ($\Delta t$ is the hourly interval, in seconds). We assume $\rho$, the density of air, to be constant (1.225 kg/m$^3$, at sea level with temperature 15 °C); the area $A$ is taken to be 1 m$^2$.

In a given year, the total number of data points, for all directions combined, is denoted by $M_a$. (This number may differ between years because of leap years or occasional gaps in the data that arise when the wind direction is too variable and hence undefined.) Thus, for a given sectorial direction $n$, the annual mean energy is

$$E_n = M_a^{-1} \sum_i E_{n,i} = \frac{1}{2} \rho A \Delta t M_a^{-1} \sum_i W_{n,i}^3 = C M_a^{-1} \sum_i W_{n,i}^3, \tag{1}$$

with constant $C = \frac{1}{2} \rho A \Delta t = 2.2 \times 10^3$ kg s/m. We follow this procedure for all individual years. Wind energy will be expressed in megajoule (MJ).

The data from weatherstation Vlieland contains a few gaps (20-21/12/2002, 13-19/2/2008, 2/7-3/8/2015). They were filled in by substituting data from another weatherstation on a neighbouring island (Terschelling Hoorn). The long-term mean energy levels are generally lower there than at Vlieland, because the station lies more sheltered, especially for northerly winds. However, for each individual wind sector the ratio is nearly constant over the years, so we can reliably fill in the missing data from Vlieland by using the data from Terschelling and applying the correction factors to the individual sectors. One day in 2002, when neither station worked, missing data was filled in by linear interpolation.

## 2.3 Wind climate and inter-annual variability

Averaging over the full 20-year record (Figure 4a), we find that the highest wind energy comes from the southwesterly and westerly directions. Together, they contain more energy than all other directions combined. The lowest energy comes from the northeasterly and southeasterly directions. We have compared this sectorial distribution with those from some other weather stations in the northeastern and southwestern corners of the Netherlands (Huibertgat en Lichteiland Goeree, respectively); they confirm this pattern, but appear less exposed to offshore winds than weather station Vlieland. In Figure 4a, we also show the difference between winter and summer half-years, again averaged over the 20-year record. For all major sectors, wind energy is substantially lower in summer, most markedly so for southerly winds.

We now examine the inter-annual variability in more detail, looking at the sectorial distribution for all individual years from 1996 to 2015 (Figure 4b). Clearly, the inter-annual variability is very large for the southwesterly direction (in terms of the absolute range of values attained), and to a lesser extent also for westerly, southerly and easterly directions. With triangles alongside the bars, we highlight the most extreme years of the record: in 1996 (dark blue), easterly winds were even more important than (south)westerly winds; conversely, in 2015 (dark red), easterly winds were at their lowest and (south)westerly winds at their highest.



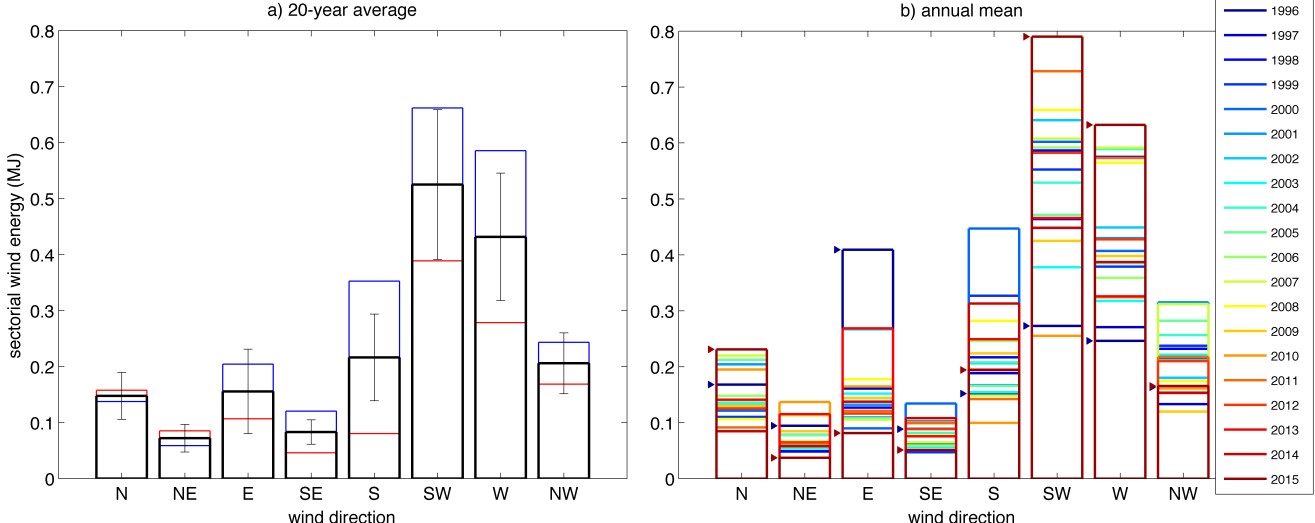

**Figure 4.** Annual mean wind energy at weather station Vlieland, divided in eight sectorial directions. In a) the average over the 20-year record (1996-2015), in black, with the typical inter-annual variability indicated in terms of the standard deviation. Also shown are the averages of the summer half-year (in red) and winter half-year (in blue). In b) the sectorial annual mean wind energy for all individual years. For clarity, the extreme years 1996 (darkblue) and 2015 (darkred) are highlighted with triangles alongside the bars.

In contrast, the annual mean wind energy for all directions combined, i.e. summed over the eight sectors, turns out to be fairly constant throughout the 20-year record, see Figure 5. In any given year, its value deviates no more than about 20% from the long-term mean, and usually much less. (The somewhat oscillating pattern, with minima in 1997, 2003 and 2010, does not appear to have any obvious connection with indices like El Niño or NAO.)

In conclusion, it is not so much the total wind energy that varies between years, but rather the share each of the directions gets from this total.

## 3 Correlations

We first give a detailed overview of the results from the tidal gauge at Den Helder before summarizing the results from the others.

### 3.1 Wind sectors and annual mean sea level

Elaborating on the results in Figure 4b for the eight wind sectors at weather station Vlieland, we calculate the correlation between the annual mean wind energy for each of the eight sectors and the annual mean sea level at Den Helder. The outcome is shown in Figure 6. The high correlations for the easterly and westerly winds stand out, with correlation coefficients $R = -0.77$ and $+0.81$, respectively. The negative coefficient means that easterly winds lower the mean sea level, as they drain off water



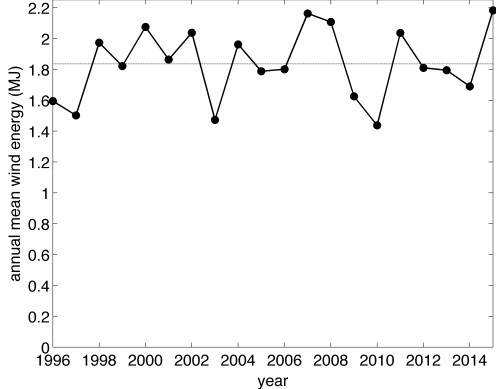

**Figure 5.** Annual mean wind energy summed over all eight sectors, at weather station Vlieland. The 20-year mean is indicated by the horizontal grey line.

from the Wadden Sea into the North Sea. Westerly winds have the opposite effect, resulting in a positive correlation coefficient. This relationship was also observed at a much shorter time scale of one tidal period (Duran-Matute et al., 2016).

The predominant role of zonal winds at the Dutch coast agrees with recent results by Dangendorf et al. (2014) and Frederikse et al. (2016) for the North Sea area. They found a contrasting outcome at the British North-Sea coast, where the role of the
wind is much smaller and is surpassed by the inverted barometer effect.

Figure 6 is useful for providing a first impression, but of course the eight sectors cannot be regarded as independent. In a vectorial sense, after all, there are only two independent components in the wind direction.

### 3.2   Vectorial wind direction, air pressure and annual mean sea level

To examine the vectorial (as opposed to: sectorial) wind direction, we return to the original wind data and now decompose
every hourly value of $D_i$ into west-east and south-north components, $D_i$ being the direction from which the wind blows. The angle $D_i$ is defined east of north.

We apply this decomposition of direction to the energy values and then sum, for every individual year, all west-east contributions and all south-north contributions:

$$E_{WE} = -CM_a^{-1} \sum_i W_i^3 \sin(D_i), \qquad E_{SN} = -CM_a^{-1} \sum_i W_i^3 \cos(D_i). \qquad (2)$$

The minus sign on the right-hand sides means that winds from the west or south count as positive, and winds from the east or north as negative. (In the hypothetical case that there is as much energy in winds from the west as from the east, the result of the summation will be zero.) To have a proper comparison between years, we divide by the total annual number of data points, $M_a$. $E_{WE}$ and $E_{SN}$ thus represent the annual mean vectorial components of the wind energy.



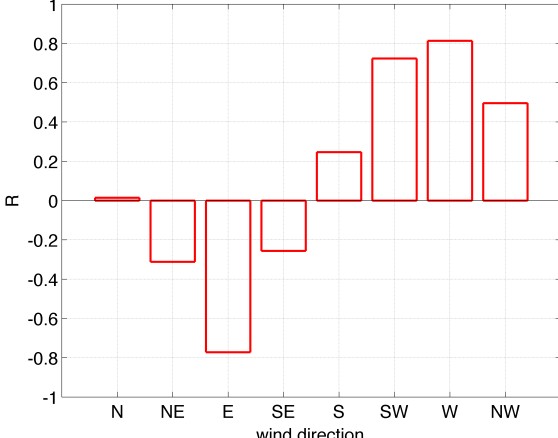

**Figure 6.** Correlation coefficient $R$, based on a 20-year record, representing the correlation between annual mean sea level at Den Helder (Figure 2) and annual aggregate wind energy for each of the eight sectors, calculated from the wind record at weather station Vlieland (Figure 4b).

### 3.3 Simple correlation

We first focus on the west-east (hereafter, WE) vectorial component *alone*, which already produces a very high correlation coefficient with annual mean sea level: $R = 0.92$. The scatterplot is shown in Figure 7. The mean WE energy is positive in all years, meaning that the westerly component dominates the easterly one, in agreement with Figure 4. However, one point in the lower left corner of Figure 7 lies close to zero; this is the anomalous year 1996, in which easterly winds were exceptionally strong and (south)westerly winds exceptionally weak. The opposite case occurred in 2015, which we correspondingly find farthest on the right in Figure 7.

The grey line in Figure 7 shows the least squares fit. Using this line and the values of the annual aggregate WE wind energy, we can calculate a reconstructed annual mean sea level, according to this fit. The result is shown in Figure 8, blue line. It lies close to the observed one (in black), which means that the WE wind already explains most of the inter-annual variability.

For the other two tide gauge stations in the Dutch Wadden Sea, we can follow the same procedure and find similar correlation coefficients: 0.90 (Harlingen) and 0.88 (Delfzijl). For the additional tide gauges shown in Figure 1, the correlation decreases down to 0.73 for Vlissingen, suggesting a weaker influence by WE winds and a larger role for other factors, which will be confirmed below.

### 3.4 Multiple regression

We can improve on this result by also including the annual mean south-north component of wind energy as well as the annual mean atmospheric pressure, $p_{ann}$. (In addition, one may include time, in years, as a fourth independent variable; we discuss



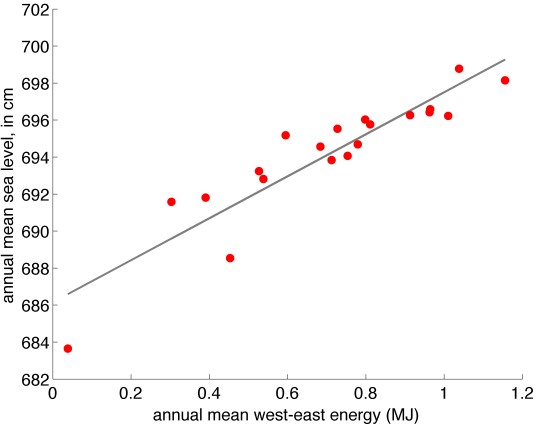

**Figure 7.** Annual mean sea level versus the annual mean west-east energy component; the correlation is $R = 0.92$. The least squares fit is shown as the grey line.

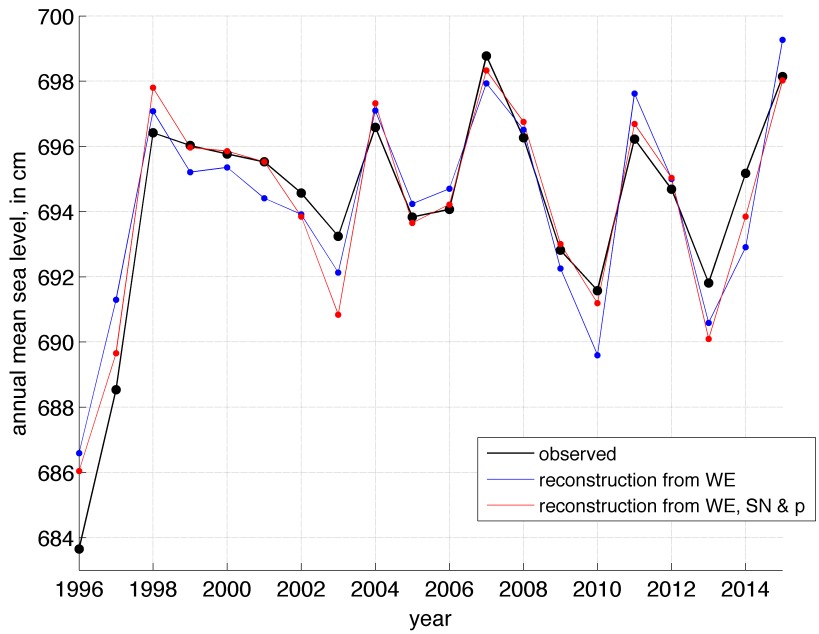

**Figure 8.** Observed annual mean sea level at Den Helder is replicated in black from Figure 2. The blue and red lines represent reconstructed annual mean sea levels based on atmospheric data. The blue line results from the annual mean west-east component of wind energy combined with the least squares fit of Figure 7. The red line uses a multiple regression involving both directions of the wind energy vector (i.e., west-east and south-north) as well as the annual mean atmospheric pressure at weather station Vlieland.





this in Section 5.1.) We deal with all three independent variables at once by using multiple regression (the backslash operator in Matlab). For Den Helder, the resulting reconstruction of annual mean sea level is shown as the red line in Figure 8.

Other stations can be treated in the same way; the results are collected in Table 2. In all cases, the west-east wind energy coefficient is dominant and positive. The south-north wind energy carries a negative coefficient, implying that southerly (northerly) winds create a lowering (surge) in mean sea level. The inverted barometer effect is found to be 0.95 cm/mbar (on average), which is close to the theoretical value of 1.0 cm/mbar (e.g., Pugh, 2004). Annual mean pressure at weather station Vlieland varies between 1013 and 1017 mbar; hence, this range accounts for an inter-annual variability in mean sea level of at most 4 cm.

| coefficient | Delfzijl | Harlingen | Den Helder | IJmuiden | Hoek van Holland | Vlissingen |
|---|---|---|---|---|---|---|
| $C_0$ ($10^3$ cm) | 1.62 (1.59) | 1.64 (1.60) | 1.48 (1.47) | 1.61 (1.60) | 1.83 (1.81) | 1.71 (1.67) |
| $C_{WE}$ (cm/MJ) | 14.9 (14.2) | 16.7 (15.5) | 12.2 (11.7) | 12.2 (12.0) | 10.0 (9.28) | 10.7 (9.57) |
| $C_{SN}$ (cm/MJ) | -2.42 (-2.55) | -1.76 (-1.97) | -3.01 (-3.08) | -3.98 (-4.02) | -5.21 (-5.34) | -7.27 (-7.47) |
| $C_p$ (cm/mbar) | -0.92 (-0.89) | -0.94 (-0.90) | -0.79 (-0.77) | -0.90 (-0.90) | -1.13 (-1.10) | -1.01 (-0.97) |
| rms error (cm) | 2.04 (1.85) | 2.00 (1.24) | 1.16 (1.05) | 1.62 (1.64) | 1.59 (1.27) | 1.95 (1.34) |
| trend (mm/yr) | 1.5±1.4 (1.6) | 2.4±1.0 (2.6) | 0.8±0.8 (0.9) | 0.4±1.2 (0.5) | 1.5±0.9 (1.6) | 2.2±1.0 (2.3) |

**Table 2.** Coefficients representing the effect of the wind climate and atmospheric pressure on annual mean sea level. Each is based on a combination of data from a tide gauge and weather station Vlieland. The root-mean-square error of the least squares fit is also listed. The trend with 95%-confidence interval is obtained from linear regression after correcting observed annual mean sea levels for meteorological effects. In brackets, we list the results from an extended multiple regression in which time is included to allow for a direct estimate of the trend; this is discussed in Section 5.1.

The upshot is that we can construct an annual mean sea level $\zeta_c$ from the atmospheric data. Overall, this constructed level corresponds with the observed annual mean sea level to within the errors listed in Table 2. The constructed level is given by

$$\zeta_c = C_0 + C_{WE} \times E_{WE} + C_{SN} \times E_{SN} + C_p \times p_{ann}.$$

The constants $C_0, C_{WE}, C_{SN}$ and $C_p$ being known from multiple regression, we can use meteorological data $E_{WE}, E_{SN}$ and $p_{ann}$ to estimate the mean sea level for any given year. It is important to realize that the constants depend on the location of the tide gauge, as indicated in Table 2. In particular, the sensitivities on WE and SN winds vary spatially. An extreme case is Harlingen, with the highest factor for WE winds and the lowest for SN winds; on the other extreme, Hoek van Holland and Vlissingen have the lowest factor for WE winds and the highest for SN winds.

If there were no wind at all, the annual mean sea level would be $\zeta_c = C_0 + C_p \times p_{ann}$. Taking the 20-year average of annual mean pressure ($\bar{p}$, 1015 mbar), we obtain a reference for mean sea level with any atmospheric variability removed.

In this analysis, we used the data from weather station Vlieland throughout. For the tide gauges like Delfzijl or Vlissingen, it would seem natural to take a more nearby weather station instead. However, it turns out that this makes the reconstruction worse. This may in part be due to the lesser quality of the data, but it also relates to the question of what spatial scale in the end



determines the local annual mean sea level. In the case at hand, presumably, the crucial factor is how the wind in the central North Sea creates surges; for this, the weather station Vlieland is more indicative than other, more remote stations.

## 3.5 Corrected levels

Finally, we can correct for the atmospheric effects as quantified in Table 2 by subtracting the atmospheric-induced variations, contained in $\zeta_c$, from the original observed annual mean sea level $\zeta_{obs}$. The result is shown in Figure 9 for the three tide-gauge stations in the Wadden Sea. We arbitrarily introduced an offset, for which we choose the reference level corresponding with long-term mean atmospheric pressure $\bar{p}$, i.e. $C_0 + C_p\bar{p}$.

Figure 9 demonstrates that the inter-annual variability is greatly reduced by the atmospheric correction. Using linear regression, we can calculate the trend of the corrected signal, along with the 95%-confidence interval on the slope; they are listed in the legend of Figure 9 and in Table 2 (for all six stations). On average, the 95%-confidence interval is ±1.1, which is smaller by about a factor of three compared to the interval derived from the original 20-year data (period 1996-2015), as discussed in Section 2.1.

Whilst the confidence intervals are still too large to draw conclusions on a possible change of trend compared to the 50-year period, their reduction demonstrates that a correction for meteorological factors offers a substantial gain.

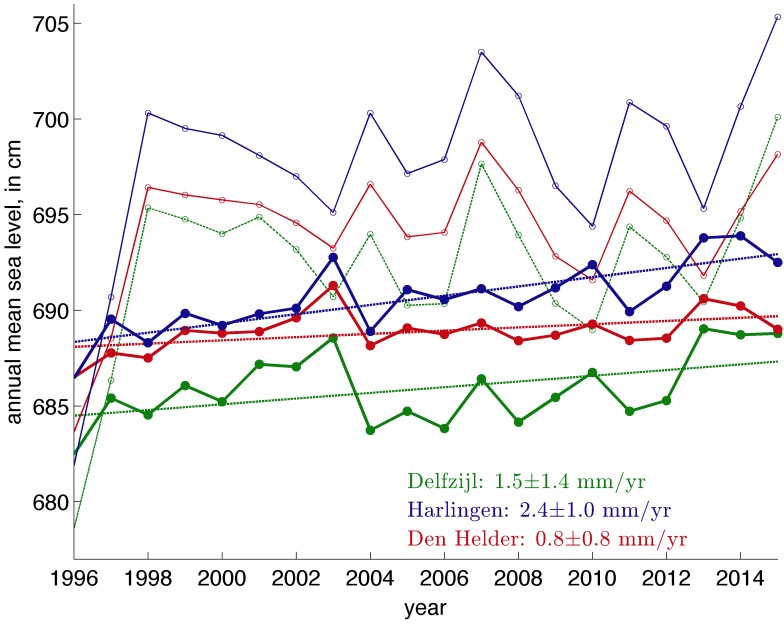

**Figure 9.** Annual mean sea level at three tide gauges in the Dutch Wadden Sea replicated in thin lines from Figure 2 for the stations Delfzijl (green), Harlingen (blue) and Den Helder (red). Also shown is the result after correction for atmospheric effects (thick lines), together with fits from linear regression (dashed). The trend with 95%-confidence interval on the slope is listed in the legend for each case.





## 4  Regional variability

Inter-annual variability of mean sea level in the Dutch Wadden Sea is evident from the three tide gauges examined in previous sections, but to identify spatial patterns in these variabilities, we need a higher spatial resolution. In Duran-Matute et al. (2014), a 3D hydrodynamic model was run for the Dutch Wadden Sea under realistic forcing, for the period 2009-2011. In this section

we explore how well the model captures the inter-annual variability, with a view to future studies, but also to identify spatial patterns. Data is used from the tide gauges shown in Figure 10a. To facilitate a comparison with the model, we will use the datum of NAP (Normaal Amsterdams Peil), which follows approximately the geoid, i.e., the equipotential of the gravity field. Water depth and bathymetry, both in observations and model, are taken with respect to this reference. In the model, however, the equipotential is represented by a plane surface.

Apart from an offset, the model captures the inter-annual variability well (Figure 10b). The offset between modeled and observed values is for each station nearly constant through the years, but it differs between stations. On average, the offset is 2.1 cm, which may be due to a slight imprecision in the open boundary conditions.

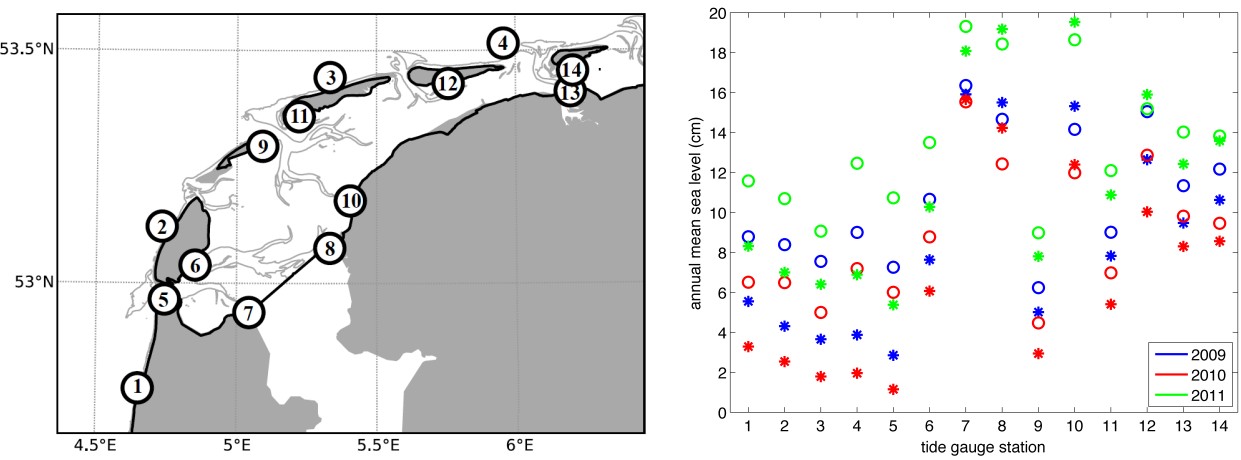

**Figure 10.** Annual mean sea level for the years 2009-2011: comparison between model results (asterisks) and observations (circles) for fourteen tide gauges in and around the Dutch Wadden Sea, whose positions are indicated in the map on the left.

Both in observations and model results, it is remarkable how strong the spatial variability is in each year, with persistently higher levels around stations 7, 8 and 10. This is further highlighted in Figure 11, a spatial plot of the 3-year mean sea level

for the period of the model run, 2009-2011. Clearly, stations 7, 8 and 10 form part of a wider area of higher mean sea levels. The location is consistent with our findings in Section 3.4: the sensitivity of mean sea level to winds in the west-east direction is strongest at Harlingen (station 10), as represented by the coefficient $C_{WE}$ in Table 2. This points to the wind as being a principal factor in the spatial variability of annual mean sea level in Figure 11. It is not that the wind itself would vary much over this small region, but rather the sensitivity of annual mean sea level to a given wind climate. In the case of this inter-

tidal area, it is plausible that this spatial variation in sensitivity is mainly determined by the morphology of the basin. Due





to the predominantly southwesterly/westerly winds (see Figure 4a), a mean wind set-up is created at eastward boundaries. Accordingly, the set-up at watersheds is generally higher at the western side than at the eastern side. More evidence of the role of the wind in the spatial variability comes from the study by Duran-Matute et al. (2016), where on tidal time scales a qualitatively similar spatial response was seen for (south)westerly winds.

Besides the wind, freshwater sources may play a role. Although we found no significant improvement in the results by adding an overall measure of freshwater discharge (as discussed below, in Section 5.4), it is conceivable that there are local effects, in particular in this case, since stations 7 and 8 lie at sluices. Future model studies (e.g. with and without freshwater discharge) could shed light on the significance of a local discharge.

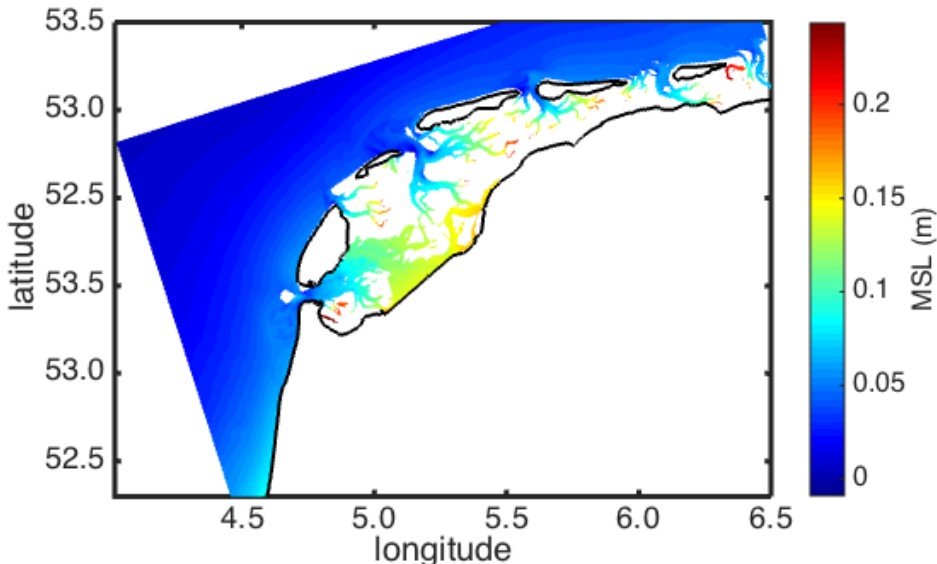

**Figure 11.** Model result: the 3-year mean sea level for the years 2009-2011, in cm. Intertidal areas, which fall dry part of the time, are left out and rendered white.

Finally, we note that inter-tidal areas offer a special challenge to determining annual mean sea level. Water level at inter-tidal
flats is strongly biased to high tide, since at low tides the flats fall dry. As the phases of low-tide are effectively non-existent on the inter-tidal flats, annual mean values of sea level would always come out too high (and indeed artificially high) in comparison with adjacent gullies. For this reason, we have left out inter-tidal areas in Figure 11 and rendered them white.

## 5   Discussion

### 5.1   Optimizing multiple regression

In the previous section we presented a way to correct for atmospheric pressure and wind effects, resulting in a corrected signal whose trend we can determine. Alternatively, we can obtain the trend directly from multiple regression by including time (in





years) as an additional independent variable, along with the wind components and pressure. This results in the values listed in brackets in Table 2. They are in fact close to those already obtained; for deriving trends, it is immaterial which procedure is followed. On the other hand, the root-mean-square errors are generally reduced by including the trend in the multiple regression, especially in the case of Harlingen, which is not surprising since the trend is strongest there.

### 5.2    Possible asymmetries in the wind effect

In the analysis in this paper, we considered the vectorial sum of the wind energy as an explanatory factor for annual mean sea level. This means that energy from westerly and easterly winds are lumped together (i.e., subtracted) as if they carry equal weight. However, it is important to keep in mind that this serves only as a first-order approach, since in reality the relation will be more asymmetric. In the case of the Dutch Wadden Sea, an easterly wind has less fetch and can drain off only a limited amount of water (because of the shallowness of the basin) compared to westerly winds, which have a longer fetch and carry a larger potential for heightening sea level, with waters coming from the large reservoir of the North Sea. Some weigh factor is thus likely to be involved, but this is beyond the scope of this paper.

### 5.3    Effect of extreme surges on annual mean sea level

An important question is whether the annual mean sea level is controlled mainly by a few extreme surges during heavy storms, or rather reflects the aggregate of all the contributions of the various conditions throughout the year. This determines whether storm surges ride, as it were, on top of a background mean sea level, or instead, they shape that very level.

To answer this question, we consider a 20-year record of the tide gauge at Den Helder (period 1996-2015, as elsewhere in this paper), with data at 10-min intervals. As a reference we take the datum NAP. During this period, mean high-tide was +59 cm, mean low-tide -80 cm. The highest level in this record is +271 cm.

We here define "extreme surges" as levels exceeding mean high-tide plus 100 cm, i.e. higher than +159 cm. With this criterion, we put the bar rather low for an event to be counted as "extreme". (This level falls in the official category "low storm surge".) Nevertheless, the combined effect of all these "exteme" events contributes on average still only +0.34 cm to the annual mean sea level, and in none of the years more than +1.0 cm.

Since the inter-annual variability of annual mean sea level lies rather in the order of several centimeters up to a few decimeters (see Figure 2), it is clear that these variations cannot be ascribed to the incidence of extreme events; instead, they must be primarily controlled by the more typical conditions that prevail in a certain year. Although intense, the extreme events last too short to leave a fingerprint on the annual mean level.

Conversely, however, there are indications that changes in mean sea level can result in a change in extremes (both in terms of level and frequency), as pointed out by Woodworth et al. (2011).



## 5.4 Other effects

Another possible cause of variability is the 18.6-year lunar nodal cycle. This cycle has two distinct effects. On the one hand, it modulates the amplitude (and phase) of the lunar constituents, notably the principal semidiurnal lunar constituent M2 and lunar declinational diurnal constituents K1 and O1. This has a very significant effect on the tidal range and on the diurnal
inequality, but it leaves annual mean sea level unaffected since high waters are as much higher as low waters are lower, giving a cancellation in the mean. On the other hand, there is a small long-period nodal constituent N, which has no effect on the tidal range but does have a signature in annual mean sea level. Exactly how important this effect is, still appears to be a matter of debate. According to Pugh (2004), the amplitude is about 4.4 mm around Europe. Baart et al. (2012) included a nodal oscillation in their fit to annual mean sea level at the Dutch coast (combining 6 tide gauge stations). They find an amplitude
of 1.2 cm, with the maximum occurring in February 2005. However, our results, after correction for atmospheric effects, show no maximum at that time. Besides, as Woodworth (2012) emphasized, the 18.6-year cycle cannot really be distinguished from decadal variability in short records (like ours), while the cycle will hardly affect trends in long ones; hence he parenthetically suggests "just forgetting it for many midlatitude coastlines" – which is what we have done in this paper.

On the decadal time scale, variability in land water storage (due to ENSO) has been found to have a fingerprint in annual
mean sea level, with a lowering during La Niña events, such as in 2007-2009 (Woodworth et al., 2011). This, however, is not clearly visible in our corrected annual mean sea levels (Figure 9).

A factor of unknown significance is the outflow of freshwater and its inter-annual variability. The results from Section 4 suggest it plays at least a role in the spatial variability in annual mean sea level. A preliminary investigation including the river Rhine discharge (as a proxy we took the discharge at Lobith, where it enters the Netherlands before splitting into different
branches), gave no substantial improvement in the multiple regression analysis, confirming a similar earlier conclusion by de Ronde et al. (2014).

## 6 Conclusions

We find that at the Dutch coast, southwesterly winds are dominant in the wind climate (Figure 4a), but west-east directions stand out as having the highest correlation with annual mean sea level (Figure 6). For different stations in the Dutch Wadden
Sea and along the coast, we find a qualitatively similar pattern, although the precise values of the correlations vary. The inter-annual variability of mean sea level can already be largely explained by the west-east component of the net wind energy vector, with some further improvement if one also includes the south-north component and annual mean atmospheric pressure.

Knowledge of these local correlations can then be used to correct values of annual mean sea for these atmospheric effects. This reduces the margin of error (expressed as 95%-confidence interval) by about a factor of three for linear trends in a 20-year
sea level record. Still, the margin is too large to draw conclusions on a possible acceleration or deceleration of sea level rise. Inclusion of other factors could perhaps reduce the error further; a possible candidate is the inclusion of annual variability in local freshwater run-off. This needs to be investigated.





We showed that a modest reduction in the margin of error of trends in mean sea level rise can be obtained by selecting the summer half-year instead of the full year because of lower wind-induced variability. However, these summer trends are not representative of the annual mean trends, in agreement with earlier findings by Dangendorf et al. (2013). For all stations studied here, we find from 100-yr long records a steeper rise in the winter half-year than in summer half-year values of mean

sea level. This is an apparently overlooked but relevant feature in view of coastal protection, since severe storms are more common during the winter half-year.

This study also implies that climatic changes in the dominant wind direction or in the strength of winds from any specific sector, may affect the annual mean sea level quite significantly. For the Netherlands, no trend in wind strength has been found for reconstructed wind fields in the 20th century (calculated as geostrophic winds from atmospheric pressure records, KNMI

(1999)), but to date, no study seems to have been carried out for possible trends in individual wind sectors. Climate studies tend to focus on extreme events such as the frequency of severe storms and maximum wind speed (see, e.g. de Winter et al., 2013) rather than on the yearly mean wind energy for different wind sectors.

Such changes in wind climate may affect locations differently; as this study shows, some places have a higher correlation with winds in the west-east (or south-north) direction than others (cf. Table 2). This sensitivity to wind direction has a very

regional variability, as isuggested by the model result in Figure 11 and supported on shorter time scales by the modelling study by Duran-Matute et al. (2016). Even on a small scale like the Dutch Wadden Sea, there is a spatial variability in the response of the annual mean sea level to changes in the wind climate.

*Competing interests.*   No competing interests

*Acknowledgements.*   We thank Thomas Frederikse and Vincent Vuik for bringing several relevant papers and reports to our attention.



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
