# Peer review of "Inter-annual variability of mean sea level and its sensitivity to wind climate in an inter-tidal basin"

_Earth System Dynamics, 2017_

## Referee Comment (RC1) · Anonymous Referee #1 · 23 May 2017

The authors studied the statistical relationship between the wind and annual mean sea level in the Dutch Wadden Sea. They found that the interannual variability of mean sea level there can be largely explained by the west-east wind component. They also reported that correcting observed values of annual mean sea level for meteorological factors can reduce the margin of error greatly. Findings from this study could help us understanding variations in the regional sea level over the Dutch Wadden Sea. However, there are a number of issues that I feel should be addressed in a revision, as detailed bellow.

Major comments: 1. The authors limited their analysis to the statistical relationship between the wind and annual mean sea level, e.g., simple correlation and regression.

However, mechanisms for their relationship are also important for people to understand regional sea level change and its possible link with climate change. I suggest the author further explore the mechanistic relationship between the wind and mean sea level in the Dutch Wadden Sea at interannual time scale, to make this study scientifically more interesting. The hydrodynamic model used in Duran-Matute et al. (2014) could be an ideal tool to investigate mechanisms for the statistical relationship reported in the present manuscript.

2. Figure 1 depicts the location of tide-gauge and weather stations. To help readers have a better understanding of the setting, it would be beneficial if the authors can plot the climatological annual mean wind and/or sea level from observation or reanalysis on top of the current map.

3. I suggest the authors describe the data sets used in the present study and methods in more details, so that readers can reproduce results in the manuscript. For example, the temporal resolution and any preprocessing of the sea level data should be provided in Section 2.1. Also, more details should be provided on how the confidence interval and the (effective) sample size is calculated.

4. Figure 5 shows the annual mean wind energy at weather station Vlieland, based on which the authors conclude that the annual mean wind energy for all directions combined stays fairly constant throughout the 20 years. I find this statement unconvincing, at least with current analysis. I suggest the authors replot their Figure 5 with a better (smaller) vertical scale. The current one is misleading. The authors further conclude that "it is not so much the total wind energy that varies between years, but rather the share each of the directions gets from this total" (Page 8, Line 5). This statement is also superficial and unsubstantiated.

5. When conducting multiple regression in Section 3.4, the authors treat the west-east and south-north winds and the annual mean atmospheric pressure as three independent variables. This can be easily examined by calculating their correlation at the

interannual (or yearly) timescale.

6. With the 20-year data, year 1996 is very different from other years and looks like an outlier in a statistical sense. I wonder how the results (statistics) change if data from 1996 is removed. This practice is useful for checking the robustness of the present findings, especially considering the limited sample size of the current data.

Minor comments: 1. There are other grammatical (e.g., Page 4, Line 9, "100-yrs record") and spelling mistakes (e.g., Page 1, Line 5, "theannual") scattered throughout the manuscript and I would suggest scanning the text more thoroughly to rectify these.

---

## Referee Comment (RC2) · Anonymous Referee #2 · 29 May 2017

The author investigated the relationship between the annual wind records from a weather station and annual mean sea level in an inter-tidal basin, the Dutch Wadden Sea. They found that the inter-annual variability of mean sea level can be largely explained by the west-east component of the net wind energy vector. By correcting observed values of annual mean level for meteorological factors, they further found the margin of error reduces by about a factor of three in the trends of the 20-year sea level record. Their results illustrated the regional variability in annual mean sea level and its inter-annual variability and highlighted the sensitivity of sea level on wind direction over the Dutch Wadden Sea.

This study is interesting and contributes to the understanding of regional climate

change. However, I have two major concerns to be addressed before I can recommend publication of the manuscript. I have detailed my comments below.

1. A mechanism is needed of winds on inter-annual variability sea level variability in this particular small region. The analysis so far is a good start, whereas the authors need to explore the physical processes and mechanism via a regional climate model.

2. Is this regional change of winds and sea level related to large-scale atmosphere and ocean changes? What is the atmosphere circulation pattern associated with the wind change over the Dutch Wadden Sea? Some effects from the westerly jet shift or ENSO?

---

## Referee Comment (RC3) · Anonymous Referee #3 · 31 May 2017

The authors present a comprehensive analysis of wind and pressure effects on mean sea level along the Dutch coastline. They use a regression approach to estimate the effects from a combination of tide gauge records and wind/pressure observations. They demonstrate that wind is the most important factor in explaining the inter-annual variability and that it strongly effects both, the long-term trend itself as well as its uncertainty. The paper is generally very well written and easy to follow, although the topic is no more particular novel. Similar results have been obtained by, for instance, Marcos and Tsimplis (2007, http://onlinelibrary.wiley.com/doi/10.1029/2007GL030641/full), Calafat et al. (2013, http://onlinelibrary.wiley.com/doi/10.1002/grl.50731/full), Dangendorf et al. (2013, 2014, 2015), and Frederikse et al. (2016a, b). What is novel here

is the use of observational records for winds and pressure. However, before making this to a particular relevant point, I would like to encourage the authors to demonstrate the added value compared to reanalysis data as in earlier works. Hence, I recommend a revision of the ms before being suitable for publication in ESD. Please find below a few more specific comments: 1. The issue of the standard error size was not first mentioned by Zervas (2009), but already discussed in Douglas (1991) and in particular acknowledged for the North Sea including Dutch stations (and the effect of an atmospheric correction) in the already referenced literature. 2. One of your foci is on the important issue of uncertainty of trend estimates. However, I am missing information on how you calculated the uncertainties. Did you calculate them based on uncorrelated noise? If yes, this needs to be corrected in a revised version of the ms. As shown by Dangendorf et al. (2015, https://www.nature.com/articles/ncomms8849) and many other studies in the recent years, sea level records show a high degree of autocorrelation, which can only be accurately accounted for by adjusting the degrees of freedom. While traditionally this has been undertaken using a simple AR1 process, Dangendorf et al. (2015) have shown that in case of the North Sea this results in a serious underestimation of the true uncertainty. Hence, I suggest that the authors should at least use an ordinary AR1 process (since they use annual data, see Bos et al. https://academic.oup.com/gji/article-lookup/doi/10.1093/gji/ggt481) in describing the uncertainty and discuss the underestimation related to that. 3. One of the novel aspects in the ms is the use of wind and pressure observations. However, I am missing a comparison to equivalents from reanalysis data to demonstrate the real effect. This could be, for instance, made by comparing the amount of explained variability for the different products. I would suggest that the observations should probably show a bit better agreement than reanalysis products. 4. Along with point three I would like to suggest making use of one the available tide+surge model outputs for the North Sea, which should give in general a proper description of the effects of atmospheric forcing. You could for instance compare your estimates using the same reanalysis wind/pressure forcing as in the model and compare the results. 5. Freshwater Discharge: The authors suggest that once of the missing components could be freshwater discharge. While I agree that this could be one particular factor, this is incomplete. The first factor, which comes into my mind, is the steric effect (including its ocean bottom pressure component), which has been estimated, for instance, by Frederikse et al. (2016, http://onlinelibrary.wiley.com/doi/10.1002/2016GL070750/pdf).

---

## Author Comment (AC3) · 7 Jul 2017

please see esd-2017-32-supplement.pdf for the response to all reviewers

Please also note the supplement to this comment:
https://www.earth-syst-dynam-discuss.net/esd-2017-32/esd-2017-32-AC3-supplement.pdf

---

## Author Comment (AC1)

We thank the referees for their constructive criticism and helpful remarks and we will address their points in a revised version as indicated below. The referees' questions are replicated in italics, followed by our response in regular font.

**Referee 1**

1) *I suggest the author further explore the mechanistic relationship between the wind and mean sea level in the Dutch Wadden Sea at interannual time scale*

We will address this point in the revision by elaborating on what we already mentioned in Section 3.1 with regard to the shorter time scales of a tidal period.  The *mechanistic* relationship between wind and surges or depressions of sea level comes into play at the time scales of *hours* and models are well able to capture this  (e.g., Zijl et al. 2013 and Duran-Matute et al. 2016). At an annual time scale, there is still a relation between mean sea level and mean wind energy (as shown by this paper and earlier studies that we mention), but we would regard this relation as statistical rather than directly mechanistic.

2) *To help readers have a better understanding of the setting, it would be beneficial if the authors can plot the climatological annual mean wind and/or sea level from observation or reanalysis on top of the current map.*

About the wind one may say that the long-term dominant wind direction is SW/W (see Fig. 4a), but to plot "the" annual mean wind or annual mean sea level is not really possible precisely because the values vary so much from year to year, as figures 4b and Fig 2 show. We will however add in Fig 1 the location of the reanalysis data that we will analyze in response to a suggestion by Referee 3.

3) *I suggest the authors describe the data sets used in the present study and methods in more details, so that readers can reproduce results in the manuscript. For example, the temporal resolution and any preprocessing of the sea level data should be provided in Section 2.1. Also, more details should be provided on how the confidence interval and the (effective) sample size is calculated.*

We will provide this information in a revised version. The sea level data was obtained from the PSMSL website as monthly mean values.  For the confidence interval, we adopted a method from the reference mentioned in our paper (Montgomery and Runger, 2003); we will provide the formula.

4) *Figure 5 shows the annual mean wind energy at weather station Vlieland, based on which the authors conclude that the annual mean wind energy for all directions combined stays fairly constant throughout the 20 years. I find this statement unconvincing, at least with current analysis. I suggest the authors replot their Figure 5 with a better (smaller) vertical scale. The current one is misleading. The authors further conclude that "it is not so much the total wind energy that varies between years, but rather the share each of the directions gets from this total" (Page 8, Line 5). This statement is also superficial and unsubstantiated.*

We will substantiate our statements in more detail based on Figs 4 and 5. Specifically, we can quantify it by calculating the relative standard deviation (standard deviation divided by the mean, and expressed as a percentage). This gives, for each of the individual directions: 28% (N), 34% (NE), 49% (E), 26% (SE), 36% (S), 25% (SW), 26% (W), 26% (NW). Doing the same for the total energy (all directions summed), we find 13%, which is much smaller than the number for any of the individual directions. These numbers bear out our statement that "it is not so much the total wind energy that varies between years, but rather the share each of the directions gets from this total "and confirm what is already visually reflected in Figs 4 and 5.

5) *When conducting multiple regression in Section 3.4, the authors treat the west-east and south-north winds and the annual mean atmospheric pressure as three independent variables. This can be easily examined by calculating their correlation at the interannual (or yearly) timescale.*

We checked this. Based on the 20-yr record, the correlation between WE and SN wind-energy levels is +0.46; between pressure and WE winds, -0.03; and between pressure and SN winds, -0.39. In the first and third cases, there is a significant, although not very high, correlation. We emphasize that the inclusion of SN winds and pressure anyway contributes little to the reconstruction of mean sea level, since it is already largely explained by the WE contribution alone.

6) *With the 20-year data, year 1996 is very different from other years and looks like an outlier in a statistical sense. I wonder how the results (statistics) change if data from 1996 is removed. This practice is useful for checking the robustness of the present findings, especially considering the limited sample size of the current data.*

The year 1996 was an exceptional year in this record because of the anomalously strong winds from the E and weak SW/W winds (as already discussed in Section 2.3). However, it is *not* an outlier in the relation between WE winds and annual mean sea level, as is demonstrated by Fig 7, where 1996 is represented by the point in the lower left corner but does not stray away from the general trend indicated by the grey line. Consequently, that line barely changes if the year 1996 is excluded.

We will correct the small typos.

**Referee 2**

We thank the referee for his/her positive appraisal that this study contributes to the understanding of regional climate change.

1) *A mechanism is needed of winds on inter-annual variability sea level variability in this particular small region. The analysis so far is a good start, whereas the authors need to explore the physical processes and mechanism via a regional climate model.*

The model used in this paper (Section 4), was already used in earlier studies to examine the effects of wind on residual circulation and sea level (Duran-Matute et al 2016, see also Duran-Matute et al 2014, both referenced in our paper). The model was forced by wind stress at the surface obtained from reanalysis data (we did not mention this but will do so in a revised version). As we emphasized already in response to Referee 1, Q1, the *direct causal mechanistic* relation between wind stress and sea level plays on the time scale of *hours* (or days at most), since the sea surface responds rapidly to this forcing. In Duran-Matute et al (2016), Fig 3a, a clear relation is therefore found in a tidal-average sense. In an annual-mean sense, a relation is still found (as shown by this paper and earlier studies that we mention), but this simply reflects the aggregate of the mechanistic connections that occur on a much shorter time scale of hours.

2) *Is this regional change of winds and sea level related to large-scale atmosphere and ocean changes? What is the atmosphere circulation pattern associated with the wind change over the Dutch Wadden Sea? Some effects from the westerly jet shift or ENSO?*

We briefly mentioned already that there is no obvious connection in our 20-yr record between total wind energy and NAO or ENSO (section 2.3), but we will in a revised version discuss in more detail what other studies have found, notably Frederikse et al. (2016), who identified a small contribution from NAO at some tide gauges.

**Referee 3**

We thank the referee for his/her appraisal of our paper as being very well written and the appreciation of the novel feature of using measured atmospheric data in this sea-level variability study.

1) *The issue of the standard error size was not first mentioned by Zervas (2009), but already discussed in Douglas (1991) and in particular acknowledged for the North Sea including Dutch stations (and the effect of an atmospheric correction) in the already referenced literature.*

Zervas (2009) conveniently summarized in a graph the standard error as a function of the length of the time series. There appears to be no similar figure or table in the paper by Douglas (1991), although he does of course mention standard errors in specific individual cases. However, we agree that it is important to add the reference to Douglas.

2) *One of your foci is on the important issue of uncertainty of trend estimates. However, I am missing information on how you calculated the uncertainties. Did you calculate them based on uncorrelated noise? If yes, this needs to be corrected in a revised version of the ms. As shown by Dangendorf et al. (2015, https://www.nature.com/articles/ncomms8849) and many other studies in the recent years, sea level records show a high degree of autocorrelation, which can only be accurately accounted for by adjusting the degrees of freedom. While traditionally this has been undertaken using a simple AR1 process, Dangendorf et al. (2015) have shown that in case of the North Sea this results in a serious underestimation of the true uncertainty. Hence, I suggest that the authors should at least use an ordinary AR1 process (since they use annual data, see Bos et al. https://academic.oup.com/gji/article-lookup/doi/10.1093/gji/ggt481) in describing the uncertainty and discuss the underestimation related to that.*

As indicated in Section 2.1, we calculate the 95% confidence interval on slope as explained in the statistics textbook by Montgomery and Runger (2003). We will provide the formula in the revised version of our paper, for the convenience of the readers. Linear regression, as we used it, is still is a very commonly applied method in the literature (see, eg, Frederikse et al. 2016). Alternative methods exist but even if they give a slightly different trend, there is no objective way to judge which is giving the "true" trend. We also note that the merits of Detrended Fluctuation Analysis are a matter of debate (Bryce and Sprague, Sci. Rep. 2, 315; DOI:10.1038/srep00315 (2012). However, we will add a comment noting that alternative methods have recently been proposed.

3) *One of the novel aspects in the ms is the use of wind and pressure observations. However, I am missing a comparison to equivalents from reanalysis data to demonstrate the real effect. This could be, for instance, made by comparing the amount of explained variability for the different products. I would suggest that the observations should probably show a bit better agreement than reanalysis products.*

We thank the reviewer for this valuable suggestion; in the revised version of this paper, we will include a comparison with reanalysis data. A first analysis suggests that the results are qualitatively similar but that the measured atmospheric data from the weather station provides more accurate reconstruction of annual mean sea level.

4) *Along with point three I would like to suggest making use of one the available tide+surge model outputs for the North Sea, which should give in general a proper description of the effects of atmospheric forcing. You could for instance compare your estimates using the same reanalysis wind/pressure forcing as in the model and compare the results.*

In Section 4, we already included model results and in a revised version we will explicitly mention that the model is in fact forced at the surface by meteorological reanalysis data. So, the comparison between modeled and measured inter-annual sea level variability already involves the use of reanalysis data.

5) *Freshwater Discharge: The authors suggest that once of the missing components could be freshwater discharge. While I agree that this could be one particular factor, this is incomplete. The first factor, which comes into my mind, is the steric effect (including its ocean bottom pressure component), which has been estimated, for instance, by Frederikse et al. (2016).*

We will rephrase the sentences on the influence of freshwater discharge to emphasize that we are considering *inter-annual* and *small-scale regional* variabilities in this paper. It would seem to us that global steric effects can barely be significant in either of these variations, because they act on longer temporal and larger spatial scales. This contrasts with local and seasonal variations of freshwater discharge.

---

## Author Response (AR1)

We thank the referees for their constructive criticism and helpful remarks and we will address their points in a revised version as indicated below. The referees' questions are replicated in italics, followed by our response in regular font.

**Referee 1**

1) *I suggest the author further explore the mechanistic relationship between the wind and mean sea level in the Dutch Wadden Sea at interannual time scale*

We address this point in the revised version by elaborating on what we already mentioned in Section 3.1 with regard to the shorter time scales of a tidal period. The *mechanistic* relationship between wind and surges or depressions of sea level comes into play at the time scales of *hours* and models are well able to capture this (e.g., Zijl et al. 2013 and Duran-Matute et al. 2016). In an annual-mean sense, a relation is still found (as shown by this paper and earlier studies that we mention), but this simply reflects the aggregate of the mechanistic connections that occur on a much shorter time scale of hours.

2) *To help readers have a better understanding of the setting, it would be beneficial if the authors can plot the climatological annual mean wind and/or sea level from observation or reanalysis on top of the current map.*

About the wind one may say that the long-term dominant wind direction is SW/W (see Fig. 4a), but to plot "the" annual mean wind or annual mean sea level is not really possible precisely because the values vary so much from year to year, as figures 4b and Fig 2 show. We have however added in Fig 1 the location of the reanalysis data that we will analyze in response to a suggestion by Referee 3.

3) *I suggest the authors describe the data sets used in the present study and methods in more details, so that readers can reproduce results in the manuscript. For example, the temporal resolution and any preprocessing of the sea level data should be provided in Section 2.1. Also, more details should be provided on how the confidence interval and the (effective) sample size is calculated.*

We have provided more details on the underlying data and methods. In Section 2.1: The sea level data was obtained from the PSMSL website as monthly mean values. For the confidence interval, we adopted a method from the reference mentioned in our paper (Montgomery and Runger, 2003); we now mention the formulas. In Section 2.2: the webpage is mentioned from which the wind data was obtained.

4) *Figure 5 shows the annual mean wind energy at weather station Vlieland, based on which the authors conclude that the annual mean wind energy for all directions combined stays fairly constant throughout the 20 years. I find this statement unconvincing, at least with current analysis. I suggest the authors replot their Figure 5 with a better (smaller) vertical scale. The current one is misleading. The authors further conclude that "it is not so much the total wind energy that varies between years, but rather the share each of the directions gets from this total" (Page 8, Line 5). This statement is also superficial and unsubstantiated.*

We have substantiated our statements in more detail In Section 2.3, based on Figs 4 and 5. Specifically, we calculated the relative standard deviation (standard deviation divided by the mean, and expressed as a percentage). This gives, for each of the individual directions: 28% (N), 34% (NE), 49% (E), 26% (SE), 36% (S), 25% (SW), 26% (W), 26% (NW). Doing the same for the total energy (all directions summed), we find 13%, which is much smaller than the number for any of the individual directions. These numbers bear out our statement that "it is not so much the total wind energy that varies between years, but rather the share each of the directions gets from this total "and confirm what is already visually reflected in Figs 4b and 5.

5) *When conducting multiple regression in Section 3.4, the authors treat the west-east and south-north winds and the annual mean atmospheric pressure as three independent variables. This can be easily examined by calculating their correlation at the interannual (or yearly)*

*timescale.*

We checked this. Based on the 20-yr record, the correlation between WE and SN wind-energy levels is +0.46; between pressure and WE winds, -0.03; and between pressure and SN winds, -0.39. In the first and third cases, there is a significant, although not very high, correlation. We emphasize that the inclusion of SN winds and pressure anyway contributes very little to the reconstruction of mean sea level, since it is already largely explained by the WE contribution alone.

6) *With the 20-year data, year 1996 is very different from other years and looks like an outlier in a statistical sense. I wonder how the results (statistics) change if data from 1996 is removed. This practice is useful for checking the robustness of the present findings, especially considering the limited sample size of the current data.*

The year 1996 was an exceptional year in this record because of the anomalously strong winds from the E and weak SW/W winds (as already discussed in Section 2.3). However, it is *not* an outlier in the relation between WE winds and annual mean sea level, as is demonstrated by Fig 7, where 1996 is represented by the point in the lower left corner but does not stray away from the general trend indicated by the grey line. Consequently, that line barely changes if the year 1996 is excluded.

We have corrected the small typos.

**Referee 2**

We thank the referee for his/her positive appraisal that this study contributes to the understanding of regional climate change.

1) *A mechanism is needed of winds on inter-annual variability sea level variability in this particular small region. The analysis so far is a good start, whereas the authors need to explore the physical processes and mechanism via a regional climate model.*

The model used in this paper (Section 4), was already used in earlier studies to examine the effects of wind on residual circulation and sea level (Duran-Matute et al 2016, see also Duran-Matute et al 2014, both referenced in our paper). The model was forced by wind stress at the surface obtained from reanalysis data (we now mention this in Section 4). As we emphasized already in response to Referee 1, Q1, the *direct causal mechanistic* relation between wind stress and sea level plays on the time scale of *hours* (or days at most), since the sea surface responds rapidly to this forcing. In Duran-Matute et al (2016), Fig 3a, a clear relation is therefore found in a tidal-average sense. In an annual-mean sense, a relation is still found (as shown by this paper and earlier studies that we mention), but this simply reflects the aggregate of the mechanistic connections that occur on a much shorter time scale of hours.

2) *Is this regional change of winds and sea level related to large-scale atmosphere and ocean changes? What is the atmosphere circulation pattern associated with the wind change over the Dutch Wadden Sea? Some effects from the westerly jet shift or ENSO?*

We briefly mentioned already that there is no obvious connection in our 20-yr record between total wind energy and NAO or ENSO (section 2.3), but we now discuss in the Introduction in more detail what other studies have found, notably Frederikse et al. (2016), who identified a small contribution from NAO at some tide gauges.

**Referee 3**

We thank the referee for his/her appraisal of our paper as being very well written and the appreciation of the novel feature of using measured atmospheric data in this sea-level variability study.

1) *The issue of the standard error size was not first mentioned by Zervas (2009), but already discussed in Douglas (1991) and in particular acknowledged for the North Sea including Dutch stations (and the effect of an atmospheric correction) in the already referenced literature.*

Zervas (2009) conveniently summarized in a graph the standard error as a function of the length of the time series. There appears to be no similar figure or table in the paper by Douglas (1991), although he does of course mention standard errors in specific individual cases. However, we agree that it is important to add the reference to Douglas and have done so in the first paragraph of the Introduction.

2) *One of your foci is on the important issue of uncertainty of trend estimates. However, I am missing information on how you calculated the uncertainties. Did you calculate them based on uncorrelated noise? If yes, this needs to be corrected in a revised version of the ms. As shown by Dangendorf et al. (2015, https://www.nature.com/articles/ncomms8849) and many other studies in the recent years, sea level records show a high degree of autocorrelation, which can only be accurately accounted for by adjusting the degrees of freedom. While traditionally this has been undertaken using a simple AR1 process, Dangendorf et al. (2015) have shown that in case of the North Sea this results in a serious underestimation of the true uncertainty. Hence, I suggest that the authors should at least use an ordinary AR1 process (since they use annual data, see Bos et al. https://academic.oup.com/gji/article-lookup/doi/10.1093/gji/ggt481) in describing the uncertainty and discuss the underestimation related to that.*

As indicated in Section 2.1, we calculate the 95% confidence interval on slope as explained in the statistics textbook by Montgomery and Runger (2003). We now provide the formula in the revised version of our paper, for the convenience of the readers (Section 2.1, second paragraph). Linear regression, as we used it, is a very commonly applied method in the recent literature (see, eg, Frederikse et al. 2016). Moreover, our 100-yr annual mean trends Table 1 are fully in agreement with other recently published results by Wahl et al. 2013 (as we mention in Section 2.1). We also note that the merits of Detrended Fluctuation Analysis appear to be a matter of debate (Bryce and Sprague, Sci. Rep. 2, 315; DOI:10.1038/srep00315 (2012). However, we added a comment (Section 2.1, second paragraph) noting that alternative methods have recently been proposed.

Finally, we note that this paper has not as its primary goal to determine trends but rather to examine the regional variability of the sensitivity of annual mean sea level to wind climate.

3) *One of the novel aspects in the ms is the use of wind and pressure observations. However, I am missing a comparison to equivalents from reanalysis data to demonstrate the real effect. This could be, for instance, made by comparing the amount of explained variability for the different products. I would suggest that the observations should probably show a bit better agreement than reanalysis products.*

We thank the reviewer for this valuable suggestion; in the revised version of this paper, we included a comparison with reanalysis data (Section 5.5). The analysis indicates that the results are qualitatively similar but that the measured atmospheric data from the weather station provides a slightly more accurate reconstruction of annual mean sea level.

4) *Along with point three I would like to suggest making use of one the available tide+surge model outputs for the North Sea, which should give in general a proper description of the effects of atmospheric forcing. You could for instance compare your estimates using the same reanalysis wind/pressure forcing as in the model and compare the results.*

In Section 4, we already included model results and in the revised version we now explicitly mention that the model was in fact forced at the surface by meteorological reanalysis data. So,

the comparison between modeled and measured inter-annual sea level variability already involves the use of reanalysis data.

5) *Freshwater Discharge: The authors suggest that once of the missing components could be freshwater discharge. While I agree that this could be one particular factor, this is incomplete. The first factor, which comes into my mind, is the steric effect (including its ocean bottom pressure component), which has been estimated, for instance, by Frederikse et al. (2016).*

We rephrased the sentences on the influence of freshwater discharge to emphasize that we are considering *inter-annual* and *small-scale regional* variabilities in this paper (Section 5.4, last paragraph). It would seem to us that global steric effects can barely be significant in either of these variations, because they act on longer temporal and larger spatial scales. This contrasts with local and seasonal variations of freshwater discharge.

---

## Author Response (AR2)

The article by Gerkema and Duran-Matute has been revised, taking into account the other reviewers' and my own comments. But, I am still not entirely satisfied with the reversion, especially the way uncertainty was calculated and discussed. Reviewer 3 shared similar concern. Sea level records show a high degree of autocorrelation, and the authors should consider adjusting the way of calculating the interval of trend. I suggest the authors use the AR1 process to calculate the effective sample size (eq 6 in Zwiers and von Storch, 1995, https://doi.org/10.1175/1520-0442(1995)008<0336:TSCIAI>2.0.CO;2) and then use the effective sample size to calculate the uncertainty (Page 3, Line 10-13). This has already been suggested by Reviewer 3, but the authors did not follow. Of course, all of their discussion of uncertainty should also be revised according to the new calculation.

*We thank the reviewer for this valuable suggestion, which we have followed, resulting in new confidence intervals in Figures 2 and 9, and in Tables 1 and 2 (last row). Corresponding changes in the text are now indicated in blue. The main conclusions still stand, but we find that a correction for atmospheric effects now reduce the confidence interval even more than in the previous calculation. Regarding the reference mentioned by the reviewer, we think that the formula has a typo (tau should be rho) – for this reason, we refer to a paper by Santer et al. (2000), where the expression is stated correctly.*